# Novel Tissue-Engineered Multimodular Hyaluronic Acid-Polylactic Acid Conduits for the Regeneration of Sciatic Nerve Defect

**DOI:** 10.3390/biomedicines10050963

**Published:** 2022-04-21

**Authors:** Fernando Gisbert Roca, Luis Gil Santos, Manuel Mata Roig, Lara Milian Medina, Cristina Martínez-Ramos, Manuel Monleón Pradas

**Affiliations:** 1Center for Biomaterials and Tissue Engineering, Universitat Politècnica de València, 46022 Valencia, Spain; nandogisbert@gmail.com (F.G.R.); lu.gils@telefonica.net (L.G.S.); cris_mr_1980@hotmail.com (C.M.-R.); 2Department of Pathology, Faculty of Medicine and Dentistry, Universitat de València, 46010 Valencia, Spain; manuel.mata@uv.es (M.M.R.); lara.milian@uv.es (L.M.M.); 3Unitat Predepartamental de Medicina, Universitat Jaume I, 12071 Castellón de la Plana, Spain; 4Biomedical Research Networking Center in Bioengineering Biomaterials and Nanomedicine (CIBER-BBN), 28029 Madrid, Spain

**Keywords:** nerve guidance conduit, hyaluronic acid, polylactic acid, aligned substrates, Schwann cells, nerve regeneration

## Abstract

The gold standard for the treatment of peripheral nerve injuries, the autograft, presents several drawbacks, and engineered constructs are currently suitable only for short gaps or small diameter nerves. Here, we study a novel tissue-engineered multimodular nerve guidance conduit for the treatment of large nerve damages based in a polylactic acid (PLA) microfibrillar structure inserted inside several co-linear hyaluronic acid (HA) conduits. The highly aligned PLA microfibers provide a topographical cue that guides axonal growth, and the HA conduits play the role of an epineurium and retain the pre-seeded auxiliary cells. The multimodular design increases the flexibility of the device. Its performance for the regeneration of a critical-size (15 mm) rabbit sciatic nerve defect was studied and, after six months, very good nerve regeneration was observed. The multimodular approach contributed to a better vascularization through the micrometrical gaps between HA conduits, and the pre-seeded Schwann cells increased axonal growth. Six months after surgery, a cross-sectional available area occupied by myelinated nerve fibers above 65% at the central and distal portions was obtained when the multimodular device with pre-seeded Schwann cells was employed. The results validate the multi-module approach for the regeneration of large nerve defects and open new possibilities for surgical solutions in this field.

## 1. Introduction

Peripheral nerve injury (PNI) is a common condition that results from trauma, laceration, stretching and/or compression of peripheral nerves. It can result in a wide range of symptoms, including severe sensor and motor deficits [1,2]. After the nerve injury occurs, the endogenous repair and regeneration process is initially characterized by a Wallerian degeneration, which involves both morphological and biochemical changes to support the new growth. The nerve regeneration begins at the proximal stump within the first 48 h, as the growing axons attempt to bridge the lesion gap and reinnervate the distal target organ [3]. However, this endogenous repair process is extremely slow, with an average rate of axonal growth of around 1 mm/day, which also cannot sustain itself for more than 12 months [4,5,6].

Treatments that provide complete functional recovery are lacking for severe PNI where the gap between the proximal and distal stumps of the injured nerve is long. If this gap is longer than 5 mm, regeneration cannot be achieved naturally [7]. Therefore, these long lesions are not capable of regenerating by themselves and require a surgical intervention [7]. Furthermore, when the injury cannot be repaired through a direct end-to-end connection without applying tension, it is necessary to use an autograft nerve from the patient himself (current standard action), a nerve allograft from a cadaver donor, or an artificial nerve conduit to bridge the gap [8,9].

The use of autograft nerves produces a morbidity of the donor site, where the formation of a neuroma can occur, in addition to the increase in cost and operative time due to having to perform a second surgery [10,11,12], and nerve allografts present potential side effects of host immunosuppression [2,9]. For these reasons, biomaterial engineered constructs, called nerve guidance conduits (NGC), are very interesting for the repair of nerve injuries [13,14,15]. Several NGC have been approved by the US Food and Drug Administration (FDA) and the European Medicines Agency (EMA) for the treatment of PNI [16,17]. However, they present several disadvantages, such as limits on the size of the nerve gap that can be repaired, poor biocompatibility and inefficient regeneration mainly due to poor revascularization of the nerve tissue. Thus, more research is needed in this field [18,19,20].

Structural cues on macro- and microscopic levels are being used to improve clinical results of NGC during the last years [21]. NGC have achieved an improvement of the guidance of regeneration, but currently only satisfactorily for short defects (gap < 15 mm) [22,23,24,25]. Conduits with multiple channels that increase the surface area of growth, the incorporation of extracellular matrix molecules and the use of supportive cells and growth factors have been shown to be more efficient compared to the use of hollow NGC [13,26,27]. However, they have not yet been able to match the regenerative capacity that is provided by autologous nerves [28].

Another factor to consider is the maximum length that can be repaired. It is still a challenge to achieve the repair of nerve injuries greater than 4 cm in length, which have limited expectations of functional recovery even with use of nerve autografts [29,30]. This can be explained by a multitude of factors, such as the limited regenerative capacity of neurons, the creation of a substrate that prevents axonal growth, the lack of trophic support, errors in axonal guidance or the reinnervation of dysfunctional targets [31,32]. Most of the repair alternatives that attempt to address some of these factors have only resulted in a modest improvement [33].

In the device that is here presented we employ two different types of biomaterials: cylinders (modules) made from hyaluronic acid (HA) and a polylactic acid (PLA) microfiber tubular bundle located in the common lumen of the co-aligned module assembly. On the one hand, HA is one of the main components of the extracellular matrix and can be crosslinked to form three-dimensional networks (hydrogels) [34,35,36]. HA presents low immunogenicity, high biocompatibility and a good biodegradability, as well as mechanical properties similar to soft nervous tissues [37,38]. Furthermore, HA hydrogels have an anti-inflammatory and inhibitory effect on astrocyte activation, thus reducing glial scar formation [39]. On the other hand, PLA is a synthetic polyester with good mechanical properties, biocompatibility and biodegradability [40,41]. PLA has a good performance for the regeneration of the nervous system, especially in the form of a fibrous scaffold [42,43,44]. The PLA microfibers located inside our HA conduits provide support for cell adhesion and migration as well as for the guidance of axonal growth [45,46]. To select the diameter of the PLA fibers, we relied on our previous experience showing that PLA microfibers with a diameter of 10 µm presented better guided axonal growth [45].

It is important to highlight that our device is not made up of a single long HA conduit, but from several short HA modules. They are kept together colinearly by the (common) internal PLA microfiber bundle and by external fixing structures at the ends of the device [47]. This multimodular approach presents several prospective advantages. It can afford longer lengths than usual unimodular concepts, providing a greater flexibility and adaptability of the device in long lesions to avoid flexural breakage. In the case of cell transplantation, this is accompanied by a better survival of the transplanted cells within the conduits, due to better fluid transport. For this very same reason, when placed in vivo, a better vascularization of the newly formed tissue inside the device is expected, thanks to the interstitial micrometrical spaces between the modules, which allow for the exchange of cells and fluids with the close neighborhood and thus the vascularization of locations far from the ends of the device. Blood vessels play a very important role in regenerating nerves to provide blood supply and nutrients, serving as pathways over which Schwann cells (SC) can migrate to form bands of Büngner that promote axonal regeneration [24,48,49,50,51]. In addition, the multi-module concept opens the possibility of differential functionalization, with different kinds of cells or bioactive molecules of each module, thus enabling a gradient of stimuli. Among all these bioactive cues, supportive cells are frequently incorporated into the lumen of NGC [23,25,52,53]. In this study. the modules have been biofunctionalized with pre-seeded human Schwann cells (hSC).

It was previously observed that SC seeded inside HA tubular conduits with a suitable inner diameter and filled with PLA microfibers were capable of proliferating and self-organizing, forming a cell sheath that covers the inside of the conduit while also covering the surface of the microfibers [54,55,56,57]. In this way, it is possible to exploit these SC structures as support for axonal growth. In addition, we have previously observed that HA conduits filled with PLA microfibers were capable of inducing preferential neuronal differentiation of progenitor cells in vitro, and presented good biocompatibility and beneficial effects in a spinal cord injury in an in vivo model [56,57].

SC are responsible for axon myelination and play a key role in regeneration and protection in the peripheral nervous system (PNS) [58,59]. Following a nerve injury in the PNS, SC contribute to nerve repair by activating, de-differentiating, dividing and proliferating distally [60]. SC act as a source of neurotrophic and angiogenic factors as well as surface proteins involved in the maintenance of normal PNS function and in the activation of immune response after an injury occurs [61]. Therefore, SC have been proposed for regenerative purposes both in the peripheral and central nervous system [62,63,64,65].

In this study, we combined SC and biomaterials to build a NGC that tries to imitate the supportive action of these cells in the natural process of regeneration. We reasoned that the presence of pre-seeded SC offered a favorable environment that could stimulate the proliferation, migration and regeneration of both neuronal and non-neuronal cell populations at the injury site, offering a more robust and efficient method for repairing critical gap nerve injuries. In addition, the highly aligned PLA microfibers would favor the guidance of cell migration and axonal growth, and the multimodular approach would entail a better vascularization of the regenerated tissue.

Here, we present a study of the performance of the multi-module implant in a rabbit sciatic nerve defect model. We initially characterized our bioengineered constructs in vitro to ensure they had the necessary properties for in vivo implantation. The animal model consisted of a critical-size (15 mm) sciatic nerve defect in the rabbit, which is an animal model that is more translational for nerve regeneration research than a rat model, and allows for testing larger NGC [66]. The effects of repair and regeneration were evaluated over a period of 6 months using several histology techniques. The primary focus of our in vivo study was to establish the benefits of both the pre-seeded SC and the multimodular conduit approach for nerve regeneration, comparing the results obtained against those for implants of one module and of a non-pre-seeded multimodular conduit.

## 2. Materials and Methods

### 2.1. Preparation of HA-PLA Conduits

Conduits of crosslinked HA from *Streptococcus equi* (1.5–1.8 MDa, Sigma-Aldrich, Madrid, Spain) were synthesized, as previously described [54,55,57] in a PTFE mould designed to obtain the desired dimensions. The conduits were hydrated in distilled water and cut to 7.5 or 15 mm length. The 7.5 mm length crosslinked hollow cylinder constitutes the unit module. Implants with one (15 mm-long) module and two (7.5 mm-long) modules were prepared for this study. A bundle of polylactic acid (PLA) microfibers was placed in the lumen of the single- or bi-module HA conduits. It was manufactured with 3600 parallel PLA microfibers with a diameter of 10 µm each (AITEX Textile Research Institute, Alicante, Spain) that were joined at their ends with an adhesive material based on cellulose. First, the microfibers were arranged parallel in various planes, and then they were held in that position by the cellulose-based adhesive. Subsequently, the flexibility of the adhesive allowed it to be rolled into a conduit, the curvature of which was maintained by adding more adhesive, forming a ring that allowed its suture to the nerve stump. Once the microfiber bundle was inserted inside of one HA module or two HA modules, the total length of the device was 20 mm, because the suture rings extended 2.5 mm at each side to introduce the nerve stumps. The final conduits were first sanitized for 2 h with 70% ethanol (Scharlab, Barcelona Spain), and then they were immersed in 50%, 30% ethanol and ultrapure water (Mili-Q^®^) for 10 min. The preconditioning of the conduits was carried out by immersion in Dulbecco’s Modified Eagle Medium with a high glucose level (4.5 g/L) (DMEM, 21331020, Life Technologies, Madrid, Spain), supplemented with 10% Fetal Bovine Serum (FBS, 10270-106/A3381E, Life Technologies, Madrid, Spain) and 1% Penicillin/Streptomycin (P/S, 15140122, Life Technologies, Madrid, Spain), and incubation took place at 37 °C for 24 h in a humidified atmosphere containing 5% CO_2_. Three groups of HA-PLA conduits were studied: a unimodular conduit (UMC) with a 15 mm-long HA conduit, a multimodular conduit (MMC) formed by two contiguous HA conduits with a length of 7.5 mm each and a multimodular conduit with human Schwann cells (hSC) pre-seeded inside the HA conduits (MMS + hSC).

### 2.2. Dimensions of HA Tubular Scaffolds and Inflation Test

Longitudinal and cross-sectional images of the HA conduits were taken with a magnifying glass (Leica, Madrid, Spain) to measure their dimensions. The images were then processed with ImageJ software to obtain the measurements. Lengths were obtained from the images of the longitudinally arranged tubes, and the measurements of the external and internal diameters were acquired from the images of the cross sections. In total, four measurements were made for each of the parameters.

To assess the water absorption capacity of HA conduits, a swelling test was carried out, studying the differences in size between the dry and swollen conduits. For this, cross sections of approximately 1 mm in length were made of the conduits, and they were introduced for 2 h in distilled water. After this time, images were taken in the magnifying glass and, with the ImageJ software, the dimensions of the external and internal diameters were measured. The swelling ratio was calculated following Equation (1), where *D* refers to the diameter, external (ext) or internal (int) of the conduits, which were measured in dry (d) and swollen (h) states.
(1)$$Swelling\;ratio=\left(D_{ext,\;s}-D_{int,s}\right)/\left(D_{ext,\;d}-D_{int,d}\right)$$

### 2.3. Human Schwann Cells Culture

Human Schwann cells (hSC, P10351, Innoprot, Bizkaia Spain) at the third cell pass were employed for cell cultures in the conduits to study cell adhesion and proliferation. After the expansion of hSC in a cell culture flask, they were washed with PBS, and a Trypsin/EDTA solution (T/E; 25200-072, Life Technologies, Madrid Spain) was added to break the cell–matrix and cell–cell interactions to remove the cells from the bottom of the culture flask. Cells were centrifugated at 1080 rpm for 5 min, and the pellet was resuspended in the Schwann cell culture medium (SCM, P60123, Innoprot, Bizkaia, Spain). The seeding with hSC was carried out for the MMC+hSC group, with a seeding density of 625,000 cells per module (the total number of cells was 1.25 × 10^6^). Then, the conduits were introduced in an incubator at 37 °C with a humid atmosphere containing 5% CO_2_ for 5 and 10 days, renewing the Schwann cell culture medium every 48 h.

### 2.4. Morphological Characterization by Field Emission Scanning Electron Microscopy

The morphology of the HA-PLA conduits was observed by means of field emission scanning electron microscopy (FESEM). Samples were cut longitudinally to obtain a complete view of the lumina of the conduit, mounted over aluminum tape and sputtered with platinum before the FESEM analysis (ULTRA 55, ZEISS, Oxford Instruments, Wiesbaden, Germany), using a voltage of 2 kV.

To study the inside of the cellular conduits (MMC + hSC) after 10 days of cell culture, the conduits were fixed in a 3.5% glutaraldehyde (GA; Electron Microscopy Sciences, Hatfield, PA, USA) solution for 1h at 37 °C, post-fixed with 1% OsO_4_ (Electron Microscopy Sciences, Hatfield, PA, USA) for 2h at room temperature and dehydrated in a series of increasingly concentrated ethanol. Then, the conduits were processed in a critical point dryer (critical point values: 328 °C, 1100 psi). The conduits were cut longitudinally to expose their internal lumina and were mounted over aluminum tape and sputtered with platinum before analysis. Samples were finally observed in a Hitachi S4800 electron microscope using a voltage of 10 kV.

### 2.5. Morphology and Adhesion of Human Schwann Cells

After 5 and 10 days of hSC culture, samples were rinsed twice in PB 0.1M and fixed in 4% paraformaldehyde (PFA; 47608, Sigma-Aldrich, Madrid, Spain) for 20 min at room temperature. After cell fixation, PFA residues were removed with 3 washes of 10 min with DPBS. Then, cells were permeabilized and blocked in DPBS with 3% bovine serum albumin (BSA; A7906, Sigma-Aldrich, Madrid, Spain) and 0.1% Tween20 (P1379, Sigma-Aldrich, Madrid, Spain) for 1 h at room temperature.

To observe the cytoskeleton of the cells, samples were incubated with Alexa Fluor™ 555 Phalloidin (A34055, Thermo Fisher Scientific, Madrid, Spain) and diluted 1:200 at room temperature for 1 h. Cell nuclei were stained with DAPI (1/1000, D9542, Sigma-Aldrich, Madrid, Spain) for 10 min. Finally, the conduits were cut longitudinally to obtain a complete view of the lumina, and a confocal microscope (LEICA TCS SP5, Leica microsystems, Madrid, Spain) was used to obtain the images.

### 2.6. Implantation of Nerve Conduits in Rabbit Sciatic Nerve Defect

The study was conducted according to the guidelines established by the European Communities Council Directive (210/63/EU) and the Spanish regulation 1201/2005. All experimental procedures were approved by the Animal Care and Use Committee of the Polytechnic University of Valencia (2019/VSC/PEA/0142). The minimal number of animals was considered for each group due to ethical concerns (*n* = 4).

Twelve male white rabbits (Oryctolagus cuniculus, three-way crossing of lines of the Polytechnic University of Valencia) aged 12 weeks and weighing 3.0–3.5 kg were randomly divided into three groups, as follows: unimodular conduit (UMC), multimodular conduit (MMC) and multimodular conduit with pre-seeded human Schwann cells (MMC + hSC) groups. The rabbits were pre-anesthetized via intramuscular injection of ketamine–xylazine (ketamine 15 mg/kg and xylazine 3 mg/kg) into the back of the neck, and then the rabbits were anesthetized via intravenous injection of Propofol (3 mg/kg for induction and 20 mg/kg/h for maintenance). The corneas of the rabbits were lubricated with a 0.2% carbomer ophthalmic gel (artificial tear) to prevent them from drying out and to prevent the appearance of corneal ulcers.

After shaving and disinfecting the area with chlorhexidine, an incision was made in the skin, taking as reference the greater trochanter of the femur and the region near the lateral condyles of the femur. A dissection was performed between the gluteus maximus and the biceps femoris that revealed the sciatic nerve. After releasing the nerve from the underlying tissue, it was irreversibly transected. A 15 mm-long segment of sciatic nerve was resected 0.5 cm proximal to the bifurcation of the nerve into the tibial and peroneal nerve branches. Then, the nerve gap (15 mm) was bridged by the unimodular or multimodular conduits with 8/0 Prolene sutures, and the wound was closed in layers using a 4/0 resorbable suture.

At the end of the surgery, immediate postoperative analgesia was administered (Butorphanol, intravenously, 0.2 mg/kg), as well as delayed postoperative analgesia 48 h after surgery (Butorphanol, intramuscularly, 0.4 mg/kg). All the animals were housed in specific pathogen-free conditions for 6 months. They moved freely and had free access to water and food. To prevent and control self-mutilation of the operated limb and the appearance of pressure ulcers during the postoperative period, rabbits wore Elizabethan collars for 1 week after surgery, the operated foot was bandaged with tape stirrups, and a softer ground was placed in the cages. The collar and the bandage were removed if there were no signs of self-mutilation, but they were applied again if signs of self-trauma were observed. The bandage was also applied again if signs of pressure ulcers were observed, after healing the affected area.

### 2.7. Histological and Morphometric Analysis

Histological evaluation of the defected nerves was performed 6 months after surgery. Our investigation did not evaluate different time points of the regenerative process; only the final time point after 6 months was evaluated. However, we thought that this was enough time to obtain a significant axonal regeneration based on previous similar studies in rabbits that demonstrate a good sciatic nerve regeneration of smaller gaps after 3 months [67,68].

Prior to the sacrifice of the animals, it was observed that the operated leg responded to the stimulus by exerting manual pressure, exerting some resistance. After anesthetizing the rabbits, the sciatic nerve containing the guidance conduit was removed from the animals, fixed in 4% PFA for 24 h and cut into three portions: proximal (approx. 3.5 mm from the proximal end of the HA conduit), central (approx. 7.5 mm from the proximal end of the HA conduit) and distal (approx. 3.5 mm from the distal end of the HA conduit). The portions were embedded in paraffin and cut transversely (5 μm thickness). Slices were stained for Hematoxylin-Eosin (HE) according to standard procedures. The analysis of the distribution and content of myelinated nerve fibers was performed using the modified Luxol fast blue method described by Sajadi et al. [69]. The cross sections of all specimens were examined by light microscope (Leica DMD 108 microscope, Leica, Madrid, Spain).

Neovascularization was determined by immunohistochemistry using a specific mouse monoclonal CD31 anti-human antibody predicted for a rabbit antibody (ab212712; Abcam; Cambridge, United Kingdom), as previously reported [70]. The sections were deparaffinized and rehydrated using a series of graded ethanol, rinsed in distilled waters and treated with 0.3% H_2_O_2_ to block endogenous peroxidase. Then, nonspecific binding was blocked by washing with a Tween 20 buffer (Fischer Scientific; Madrid, Spain). Antigens were retrieved by boiling in a pressure cooker for 3 min in a high-pH Envision FLEX Target Retrieval Solution (Dako; Barcelona, Spain). Samples were incubated overnight with the primary antibody at 4 °C using 1:250 dilution in Envision FLEX antibody diluent. After washing with PBS, the secondary antibody (goat anti-mouse IgG-HRP, 1:200 dilution) was incubated at RT for 2 h, and then developed using the 3,3′-diaminobenzidine chromogen (Dako; Barcelona, Spain) according to the manufacturer’s instructions, which resulted in brown staining in the immunoreactive structures. Finally, the sections were counterstained with Mayer’s hematoxylin (Merk; Kenilworth, NJ, USA).

Morphometric analysis was carried out in the central and distal sections of each of the animals included using the Image Pro Plus 7.0 software (Media Cybernetics, Rockville, MD, USA). Collagen and myelinated nerve fibers were measured in five different fields of each of the Luxol fast blue-stained sections (Figure 1). The areas of the nerve fibers and collagen were normalized to the available area once the area of the PLA microfibers, vessels and cavities was subtracted, in order to compare samples with different available areas. The central and distal sections of each of the animals included were considered. CD31+ vessel density was measured by counting the number of immunostained vessels in five different fields.

### 2.8. Statistical Analysis

The data are expressed as the mean ± SEM, and GraphPad Prism^®^ software was employed for the statistical analysis. The two-way ANOVA test, together with a multiple sample mean comparison (Tuckey’s multiple comparisons test with a significance degree of 95%), was used to reveal significant differences between conditions. Statistically significant differences are indicated by *, **, *** or ****, indicating a *p*-value below 0.05, 0.01, 0.001 or 0.0001, respectively.

## 3. Results and Discussion

### 3.1. Fabrication and Characterization of HA-PLA Conduits

Three different types of nerve guidance conduits were studied, consisting of a 3600 PLA microfiber bundle with a cylindrical arrangement within one long or two short HA conduits (Figure 2). In the UMC (unimodular conduit) group, the microfiber bundle was inserted into a single HA conduit with a length of 15 mm. In the MMC (multimodular conduit) group, the microfiber bundle was inserted into two consecutive HA conduits with a length of 7.5 mm each, reaching a total length of 15 mm. Finally, in the group MMC + hSC (multimodular conduit + human Schwann cells), the MMC was pre-seeded with 1.25 × 10^6^ hSC that were cultured during 10 days prior to the intervention to obtain a continuous coverage of the PLA microfibers and the lumen of HA conduits, as explained in Section 3.2. At both extremes of the PLA microfiber bundle, a 2.5 mm-long fastener ring maintains the structure of the microfibers and allows the suture to the nerve stump that is inserted into it. The three types of implants were employed to bridge a 15 mm-long defect in a rabbit sciatic nerve model. These experimental groups allow for observing the effect of the unimodular approach versus the multimodular approach (UMC vs. MMC groups) and the effect of pre-seeded Schwann cells (MMC vs MMC + hSC groups). The inclusion of the UMC + hSC group would have provided little additional information and was not included in the study, following the reduction principle in animal experimentation.

For the morphological characterization of the HA-PLA implants, FESEM images of the HA conduits and the PLA microfiber bundle were obtained (Figure 3). As can be observed in Figure 3C, the HA conduit walls present a porous microscopic structure that allows for the diffusion of water and small molecules. The lumen’s inner surface of the conduits has a much smaller porosity (Figure 3D) that impedes cell migration and helps keep the seeded cells inside the conduit. The dimensions of the dry HA conduits were 3.0 ± 0.1 mm in internal diameter and 6.0 ± 0.2 mm in external diameter. The HA conduits had a great capacity to absorb water, and the dimensions of the swollen HA conduits were 5.9 ± 0.1 mm in internal diameter and 10.1 ± 0.6 mm in external diameter, with a swelling ratio of 1.4 ± 0.3. The diameter of the rabbit sciatic nerve is comprised between 2 and 4 mm [66,71]. Thus, the internal diameter of HA conduits in the swollen state is big enough to avoid a possible compression of the nerve. In addition, some space is necessary for the PLA microfiber bundle that is placed inside the HA conduits, whose ends are the ones that adapt and are sutured to the nerve stumps.

HA presents groups capable of forming hydrogen bonds with water, such as -OH and -NH_2_, thanks to which the HA can absorb large amounts of water. The amount of water absorbed can be diminished with the crosslinking degree [72] and, in this way, the dimensional change due to swelling can be adjusted to the size of the injured nerves. The high water content of HA in its swollen state is critical for the non-adherent properties of the HA conduit, which prevents undesired adherences to the surrounding tissues [73]. Many pharmaceutical agents such as Doxorubicin have been tried to prevent nerve scarring and adhesion in different experimental studies, but results have been poor. Therefore, the antiadherent action of HA cannot be replaced by drugs at the moment [73]. In addition, other beneficial effects of HA associated with nerve regeneration (number of axons, nerve fiber diameter and myelin thickness) have also been reported [73].

In Figure 3E, the end of the PLA microfiber bundle can be observed, with the flat ring used to keep the cylindrical arrangement of PLA microfibers marked with asterisks. At the same time, this ring represents a convenient surface for the suture of the nerve stumps to the microfiber bundle. As can be observed in Figure 3F, the PLA microfibers present a highly aligned distribution, which is necessary to obtain a high directionality of the axons, as has been observed in previous studies [45,46].

### 3.2. Growth and Distribution of Human Schwann Cells in HA-PLA Conduits

Schwann cells are one of the main cells of the peripheral nervous system, and if they are pre-seeded into a nerve guidance conduit, they can promote the growth of neurites by the secretion of neurotrophic and angiogenic factors [58,59,60,61]. To assess the behavior of hSC seeded on top of the PLA microfiber bundle, confocal and FESEM images of the cells were taken after the stablished culture time (1.25 × 10^6^ hSC cultured for 10 days). As can be observed in Figure 4A, hSC form a continuous multilayer of cells that cover all the PLA microfibers, even bridging the spaces between microfibers. Figure 4B,C show FESEM images of the hSC attached and growing on top of PLA microfibers, wrapping them. In Figure 4C, it can be clearly appreciated how the cytoskeleton of hSC is elongated in the direction of the PLA microfibers, leading to an alignment of the cells. This topographical guidance of PLA microfibers will also guide the growth of axons in the direction of the microfibers, as previously observed [45,46]. Therefore, the well-distributed hSC on the whole PLA microfiber bundle provide a convenient microenvironment for the subsequent axonal extension during the in vivo nerve regeneration process.

Besides embracing the PLA microfibers, the pre-seeded SC form a continuous planar cell sheath of a few cells’ thickness that coats the inner surface of the lumina of the HA conduits, as can be observed in Appendix A and Figure 5. To compare differential SC coverage when using a UMC or a MMC, $2.5\times 10^{6}$ and $1.25\times 10^{6}$ cells were seeded on both conduits and cultured for 5 and 10 days, respectively. As can be observed in Appendix A, when a UMC was employed, an incipient SC sheath was visible when $2.5\times 10^{6}$ cells were cultured for 5 days (Appendix A). When $1.25\times 10^{6}$ cells were cultured for 10 days, (Appendix A) a more developed SC sheath was observed, but it was not enough to cover all the lumen of the HA conduit. In the case of the MMC (Figure 5), when $2.5\times 10^{6}$ cells were cultured for 5 days (Figure 5A–C) a clearly thicker and larger SC sheath was obtained, but some bald spots could still be observed. However, when $1.25\times 10^{6}$ cells were cultured for 10 days (Figure 5D–F), a thick and continuous SC sheath was clearly observed. These results proved that it was better to culture $1.25\times 10^{6}$ cells for 10 days than to culture $2.5\times 10^{6}$ cells for 5 days in both cases. In addition, a thicker and more continuous SC sheath was obtained when using a MMC than a UMC, probably a consequence of a better distribution of the seeded cells in the case of the shorter modules of the MMC sample than in the longer UMC module. The multimodular design entails a more uniform distribution of the pre-seeded Schwann cells because it facilitates the seeding process: the cells are seeded at both ends of each module, so a shorter module reduces the distance between the seeding zones. For this reason, a better coverage of the lumen of the conduit is achieved. In addition, the use of shorter modules allows a better exchange of nutrients than the use of a longer module, whose center has fewer nutrients available [52,74,75]. This can also affect the cell distribution in the central part of the lumen of the conduit.

### 3.3. In Vivo Sciatic Nerve Regeneration

Unimodular and multimodular versions of our construct were implanted into a 15 mm-long rabbit sciatic nerve defect for 6 months. Three different groups were studied (Figure 2): a unimodular conduit (UMC) with a single 15-mm long HA module, a multimodular conduit (MMC) with two 7.5-mm long HA modules and a multimodular conduit with pre-seeded hSC (MMC + hSC). The total length of the conduits was 20 mm, including the 2.5 mm-long end bands used to keep the microfibril bundle in place and to suture the constructs to the nerve stumps (Figure 6).

Hematoxylin-eosin staining (Figure 7) shows the HA conduit (marked with *) enclosing the PLA microfibers (marked with arrows), which are found inside the lumen of the HA conduit. The PLA microfibers are homogeneously distributed, covering a large cross-sectional area of the lumen. The HA conduit presents quite a preserved structure, with only an incipient degradation 6 months after its implantation (Figure 7A–C,G–I), so it will retain its structure long enough for nerve regeneration to complete before being replaced by new tissue. The incipient degradation of the HA conduit is based on the detailed visualization of the histological images, where the HA porous structure presents voids and breaks in continuity for all the experimental groups (marked with circles in Figure 7 and Figure 8). The same is presumed to happen to the PLA microfibers, but in longer term this is due to their slower degradation rate (median half-life around 30 weeks) [76]. Until its complete degradation, the HA conduit has protected the PLA microfibers inside from possible adhesions with the surrounding tissue, maintaining the disposition of the PLA microfibers. In addition, the HA conduit has served to maintain the structure of the hSC sheath previously cultivated inside its lumen for the MMC + hSC group.

Commonly used NGC for the regeneration of the peripheral nervous system made of type I collagen, polyglycolic acid or porcine small intestinal submucosa degrade after 3 to 4 months [16]. By contrast, the HA conduit in our experiments lasted for 6 months with quite a preserved structure. This may represent an advantage, so that the HA conduit can fulfill its function during the entire time that the nerve regenerates, which takes several months. Since it will, nonetheless, eventually be resorbed, the HA hydrogel conduit is preferable to non-biodegradable NGC employed in the clinic made of materials like polyvinyl alcohol or silicon, whose destiny is to remain undegraded forever, usually encapsulated [16].

In all the studied specimens, the presence of a newly formed connective tissue was observed, which was well-organized, with collagen fibers (mainly type I collagen) and fibroblasts, along with a variable number of nerve fibers (Figure 7). The presence of an inflammatory response was not detected at 6 months. No aggregates of lymphocytes or PMNs were observed in any of the samples studied. In the same way, no adipose infiltrate was observed. Other normal cell types of the conjunctiva, mainly fibroblasts, were also observed. Inflammatory infiltrates of lymphocytes, neutrophils, eosinophils, or plasma cells were not observed. This is indicative of the absence of inflammation.

The PLA microfibers appeared quite separated, occupying much of the lumen of the HA conduit. The presence of PLA microfibers coincided with the presence of a neo-formed well-configured and balanced connective tissue where the presence of nerve fibers was appreciated (Figure 7D–F,J–L). Macrophages and giant cells were observed associated with PLA microfibers, embracing them where they appear as giant foreign body cells (Figure 8D–F). Macrophages are related to the resorption of the material and the regeneration of connective tissue, given their ability to eliminate detritus of the lesion and to control the functioning of fibroblasts. For the UMC and MMC groups (Figure 8G,H), some void spaces were observed between the PLA microfibers and myelinated nerve fibers. However, for the MMC + hSC group (Figure 8I), the myelinated nerve fibers were much closer to the PLA microfibers, embracing them completely. We think that this is due to the presence of pre-seeded Schwann cells on the surface of the PLA microfibers, which associate with the nerve fibers, causing the nerve fibers to remain much closer to the PLA microfibers.

Modified Luxol fast blue method was employed to characterize the presence and distribution of myelinated nerve fibers [69], and morphometric studies were carried out. Since we assume a good regeneration in the proximal portion with respect to the lesion, we focused on the analysis of the central and distal groups. In general, we were able to detect the presence of a high amount of myelinated nerve fibers in all the experimental groups, with a similar content of myelinated nerve fibers at the central and distal portions.

In the central portion, the area occupied by myelinated nerve fibers from the total available area was higher in the MMC group than in the UMC group (Figure 9). For the UMC group, a content in myelinated nerve fibers of 28 ± 7% was measured, whereas for the MMC group, it increased to 47 ± 14% (Figure 9D). Regarding the animals with two modules and pre-seeded hSC (MMC + hSC group), an additional increase in the area occupied by myelinated nerve fibers was observed, arriving at 69 ± 9%, confirming the beneficial effect of pre-seeded hSC. This trend was maintained in the distal portions of the scaffolds studied, with 27 ± 9%, 45 ± 14%, and 65 ± 5% of the area occupied by myelinated nerve fibers for the UMC, MMC and MMC + hSC groups, respectively. The amount of collagen presented an opposite trend, where groups with a higher amount of myelinated nerve fibers presented a lower amount of collagen, and vice versa (Figure 9E).

Several studies have evaluated the in vivo regeneration of the sciatic nerve after a crush injury, correlating the histological findings with the functional and electrophysiological recovery [77,78]. It was observed that 2 weeks after injury, almost all axons presented a Wallerian degeneration, a decomposition of the myelinated structure. This implied that, after 2 weeks, the number of axons and the area of myelinated axons significantly decreased compared with week 0, without an observable motor function recovery. The regenerating nerve fibers appeared histologically 4 weeks post-injury, but the motor function recovery remained missing. At 6 weeks after injury, the area of myelinated axons increased significantly to 30%, correlating to an increased motor function recovery. At 8 weeks after injury, the area of myelinated axons had an additional increased 50%, correlating to a good motor function recovery, scoring 4/4 in the toe-spreading index (TSI) for behavioral analyses. In our case, at 6 months after injury, the area of myelinated nerve fibers was above 60% in both the central and distal segments, which is a value correlated to a total motor function recovery [77,78].

Different studies have also evaluated the correlation between the histological and electrophysiological outcomes after the full section of the sciatic nerve, bridging the gap with a NGC [25,67]. An equivalent performance of both techniques was observed, and when the histological images showed regenerated nerve fibers in the distal stump, the electrophysiological analysis showed the functional reinnervation of downstream muscles, with a direct correlation between the number of regenerated nerve fibers and the compound muscle action potential (CAMP) amplitude of the gastrocnemius muscle [25].

One of the greatest challenges in the implantation of long NGC in the peripheral nervous system is a lack of oxygen and nutrient diffusion, specially at the central zone of the graft, which prevents the migration of SC and axons. After the guidance conduit is implanted, the growth of capillaries in the central part of the lumen may be too slow to provide adequate nutrients to the cells, inhibiting tissue formation in the scaffold core. This is due to the slow neovascularization process of the regenerated nerve, which starts from both ends of the implant. Therefore, achieving a successful vascularization is one of the main objectives in nervous tissue engineering. The multimodular design of our NGC was presumed to favor the vascularization of the neo-formed tissue, since the inter-module gap represents a way for cell migration at the midpoint of the implant’s length.

To study this process, immunohistochemistry for CD31 was used. To avoid artifacts derived from the immunohistochemical technique, all the studied sections were stained at the same time. As shown in Figure 10, the number of CD31 positive vessels was higher in the central and distal portions of the MMC (Figure 10C,D) and MMC + hSC groups (Figure 10E,F) when compared to the UMC group (Figure 10A,B). For the central portion, 12 ± 2 vessels/mm^2^ were counted for the UMC group, and 39 ± 3 vessels/mm^2^ and 34 ± 3 vessels/mm^2^ were counted for the MMC and MMC + hSC, respectively (Figure 10G). The same trend was observed for the distal portions, with 20 ± 1 vessels/mm^2^ for the UMC group, 47 ± 5 vessels/mm^2^ for the MMC group and 47 ± 11 vessels/mm^2^ for the MMC + hSC group (Figure 10G). Therefore, the MMC groups presented more microvessels, which correlates with an increased expression of angiogenic markers, as reported previously [79].

The higher vascularization of the MMC and MMC + hSC groups when compared with the UMC group may be traced to the gap (around 100 µm) between the two HA modules of the implant. This inter-module gap improves the oxygen and nutrients exchange of the central portion of the guidance conduit, leading to an improved vascularization of this zone and, therefore, to a more efficient nerve regeneration.

An upgrade that could be incorporated into the concept of our implant is a prior seeding of the modules with HUVEC, which can give rise to a pre-vascularization inside the conduit [80]. This pre-vascularization of scaffolds can improve the construct maturity and the blood perfusion after its implantation in vivo [81,82]. Long NGC require an adequate blood supply throughout their entire length, and the central zone usually lacks it. An early vascularization of the whole length of the guidance conduit could improve nutrient availability for Schwann cells and contribute to the growth of new axons.

Several limitations of this study should be considered. Firstly, the study would have benefited from the inclusion of more rabbits in each surgical group, but it must be considered that it is a pilot trial to assess which of the three studied groups achieved a better nerve regeneration, and the number of animals had to be small. Future studies will increase the number of animals employing the MMC + hSC group, which was the better one. Secondly, it should be noted that the demonstration of axonal regeneration through the injured nerve does not necessarily translate to the recovery of the function. The path of the axon until its arrival to the final organ is one of the most interesting phases of the regeneration process, but its arrival alone does not guarantee the return of useful function [83]. Therefore, new electrodiagnostic studies are required to reveal conduction through our NGC.

## 4. Conclusions

A multimodular concept for a nerve guidance conduit based on a PLA microfiber bundle inside HA conduits has been manufactured and studied in vivo in a long-defect lesion model. The inner diameter and length of the elements were adapted to meet the size requirements of a long-gap (15 mm) rabbit sciatic nerve defect. Schwann cells supporting axonal growth were seeded into the implant and cultured prior to surgery. The device provides a highly flexible structure with an aligned microfibrillar core inside hollow hydrogel modules that serves as a substrate for cell colonization, thus supporting axonal growth in a directional manner. The HA cylinders prevent adherences to the surrounding tissues, retain the pre-seeded auxiliary Schwann cells in place and protect the regeneration process from outer inflammatory cells. The multimodular design facilitates a more uniform distribution of pre-seeded cells in the conduit than in the case of a single, equal-length module. The multimodularity of the concept allows for achieving greater total lengths by placing different numbers of modules consecutively. Implanted for 6 months in a sciatic nerve defect model in rabbits, the device enabled the regeneration of the nerve, with a high percentage of myelinated fibers reaching the central and distal portions of the length. The multimodular design achieved a better neovascularization than the unimodular design and led to a better regeneration. The pre-seeded Schwann proved to be beneficial for the regeneration process since an additional increment of the quantity of myelinated nerve fibers was observed. Thus, the group that combined the multimodular design with the pre-seeded Schwann cells had the highest nerve regeneration, accompanied by a higher vascularization. These results validate the combination of multimodularity and cell supply as a means to increase the efficiency of nerve regeneration.

## 5. Patents

M. Monleón Pradas, C. Martínez Ramos, L. Rodríguez Doblado, F. Gisbert Roca. Dispositivo modular para regeneración nerviosa y procedimiento de fabricación. ES20210030065 20210127, ES2818424, 2021.

## Figures and Tables

**Figure 1 biomedicines-10-00963-f001:**
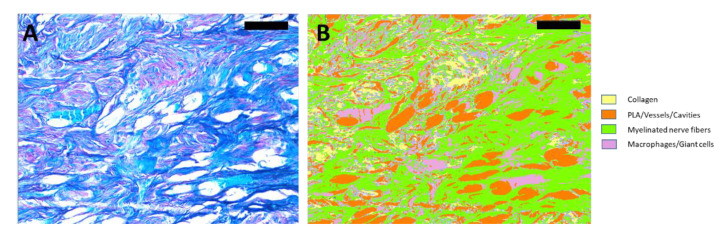
Example of a Luxol fast blue image and its segmentation. (**A**) Luxol fast blue image. (**B**) Segmentation of image (**A**). As can be observed with this example, Luxol fast blue images were segmented to quantify the area of collagen, PLA/vessels/cavities, myelinated nerve fibers and macrophages/giant cells. Collagen (red) is segmented in yellow, PLA/vessels/cavities (white) are segmented in orange, myelinated nerve fibers (dark blue) are segmented in green, and macrophages/giant cells (light blue) are segmented in purple. Scale bar = 100 µm.

**Figure 2 biomedicines-10-00963-f002:**
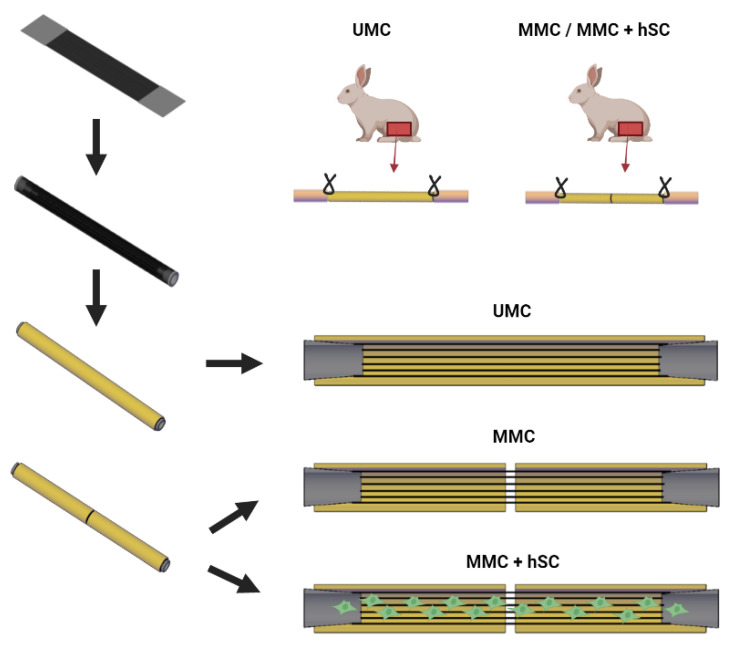
Scheme representing the differences among the three different guidance conduits that are studied. For all groups, the starting point is a PLA planar microfiber bundle formed by 3600 parallel PLA microfibers with a diameter of 10 microns each. The bundle is rolled up to obtain a cylindrical bundle. For the unimodular conduit (UMC) group, the PLA microfiber bundle is inserted into a 15 mm-long HA conduit. For the multimodular conduit (MMC), the PLA microfiber bundle is inserted into two consecutive HA conduits with a length of 7.5 mm each. The multimodular conduit plus human Schwann cells (MMC + hSC) is obtained in the same way as the MMC, but it was pre-seeded with 1.25 × 10^6^ human Schwann cells (hSC) 10 days before the implantation. At both extremes of the conduits, a 2.5 mm-long ring maintains the structure of PLA microfibers and allows the suture to the nerve stump that is inserted into it. The total length of the implant is of 20 mm. All three types of implants are used to bridge a 15 mm-long sciatic nerve defect in rabbits. Picture created with BioRender.com.

**Figure 3 biomedicines-10-00963-f003:**
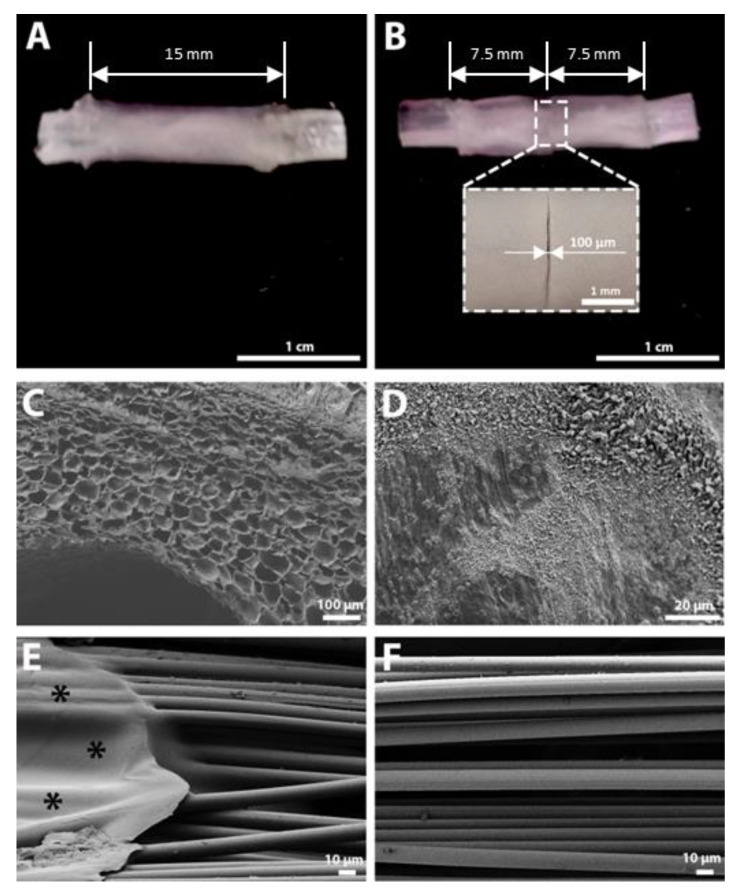
Macroscopic and field emission scanning electron microscope (FESEM) images of the acellular conduits. (**A**) Macroscopic image of the unimodular conduit (UMC). (**B**) Macroscopic image of the multimodular conduit (MMC) incorporating a detail of the central zone of the conduit to appreciate the gap between both HA modules. An arrow indicates the inter-module separation. (**C**) FESEM image of a cross section of the hyaluronic acid (HA) conduit showing the porous structure. (**D**) FESEM image of the HA conduit inner surface. (**E**) FESEM image of an end of the polylactic acid (PLA) microfiber bundle showing the flat ring (marked with *) that holds microfibers in position and allows the suture of the conduit to the nerve stump. (**F**) FESEM image of the PLA microfiber bundle showing the high alignment of PLA microfibers.

**Figure 4 biomedicines-10-00963-f004:**
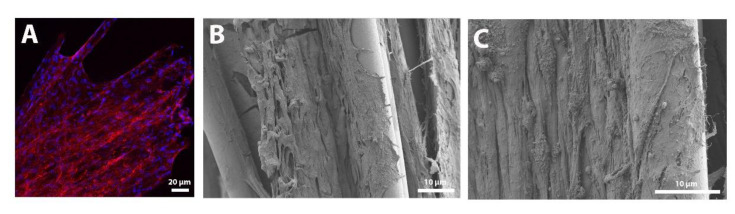
Confocal and field emission scanning electron microscope (FESEM) images of the cellular conduits. (**A**) Confocal image of human Schwann cells (hSC) seeded on the polylactic acid (PLA) microfiber bundle. The cytoskeleton of hSC is observed in red (phalloidin) and the nuclei of cells is observed in blue (DAPI). (**B**,**C**) FESEM images of hSC seeded on the PLA microfiber bundle. It can be observed how hSC can attach and grow on PLA microfibers, wrapping them (**B**). Further, an alignment of hSC cytoskeleton is observed, thanks to the topographical guidance of PLA microfibers (**C**).

**Figure 5 biomedicines-10-00963-f005:**
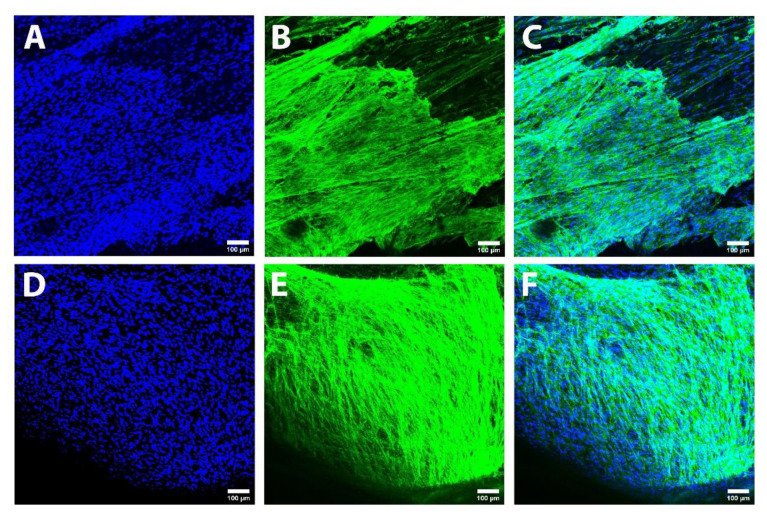
Confocal images of human Schwann cells (hSC) seeded on a multimodular conduit (MMC). (**A**–**C**) Images after the culture of $2.5\times 10^{6}$ hSC for 5 days. (**D**–**F**) Images after the culture of $1.25\times 10^{6}$ hSC for 10 days. The cytoskeleton of hSC is observed in green color (Phalloidin) and the nuclei of hSC is observed in blue color (DAPI). As can be seen, the cells have formed a sheath that fills the lumen of the HA conduit and that also covers the surface of PLA microfibers. A more continuous and thicker cell sheath is observed when $1.25\times 10^{6}$ hSC are cultured for 10 days (**D**–**F**). Further, the higher degree of coating by hSC can be appreciated when using MMC instead of UMC (Appendix A).

**Figure 6 biomedicines-10-00963-f006:**
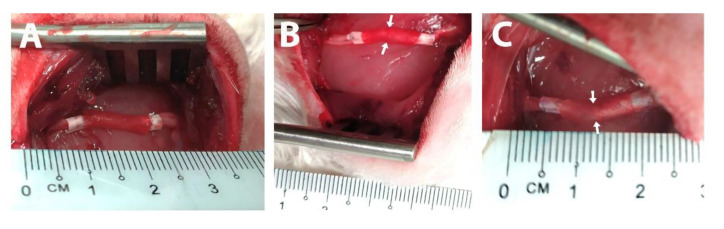
Photographs of the surgery after the insertion of the implants. (**A**) Unimodular conduit (UMC). (**B**) Multimodular conduit (MMC). (**C**) Multimodular conduit with pre-seeded human Schwann cells (MMC + hSC). In white are the end bands employed for the insertion and suture of the distal and proximal nerve stumps. The total length of the implants (one conduit for the UMC group and two conduits for the MMC and MMC + hSC groups) was 20 mm, including the 2.5 mm-long end bands. The length of the nerve gap was 15 mm. The arrows in B and C indicate the separation between both HA modules.

**Figure 7 biomedicines-10-00963-f007:**
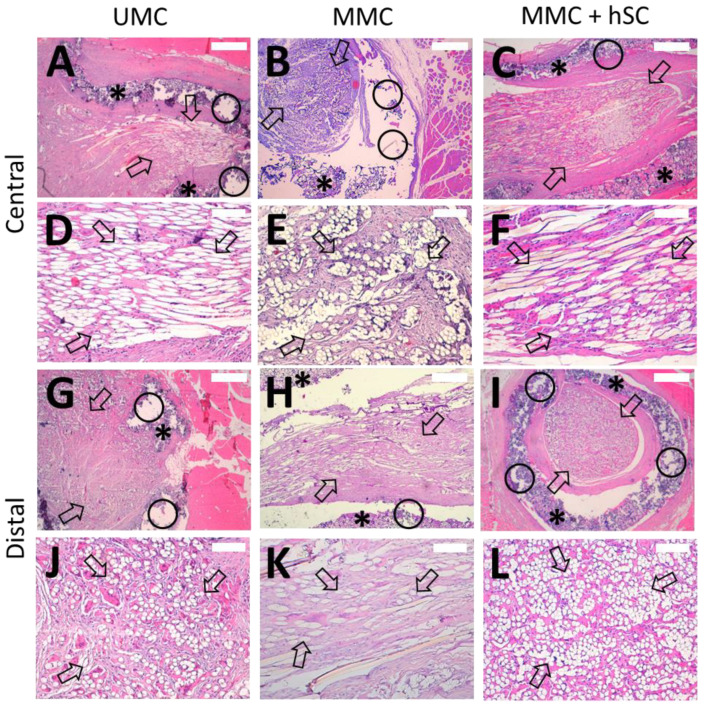
Hematoxylin and eosin staining of tissue samples from the three experimental groups 6 months after surgery. Representative images of central portions of UMC (**A**,**D**), MMC (**B**,**E**) and MMC + hSC (**C**,**F**) groups as well as distal portions of UMC (**G**,**J**), MMC (**H**,**K**) and MMC + hSC (**I**,**L**) groups are shown, with magnifications of 4× (**A**–**C**, **G**–**I**) and 20× (**D**–**F**, **J**–**L**). Cross sections of the HA conduit (marked with *) can be observed enclosing the PLA microfiber bundle (marked with arrows), which is placed inside the lumen of the HA conduit. The HA conduit walls present signs of incipient degradation in all experimental groups, since their porous structure presents voids and breaks in continuity (marked with circles). The PLA microfibers appear quite separated, occupying much of the lumen of the HA conduit to support and guide axonal growth. A well-organized connective tissue can be observed, with the presence of collagen and nerve fibers. The presence of an inflammatory response was not detected. Scale bar = 500 µm for 4× images, 100 µm for 20× images.

**Figure 8 biomedicines-10-00963-f008:**
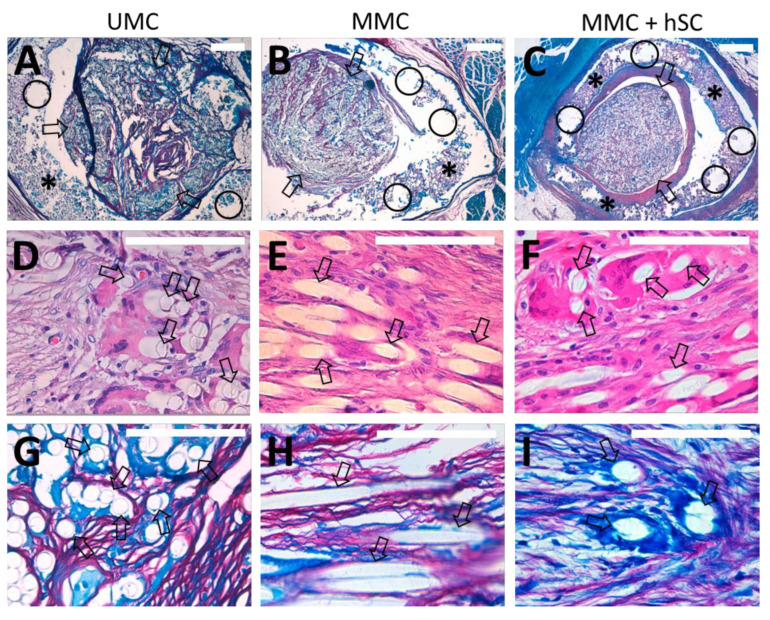
Luxol fast blue staining of tissue samples from the three experimental groups 6 months after surgery at the distal portion (**A**–**C**) and details of hematoxylin and eosin staining (**D**–**F**), as well as Luxol fast blue (**G**–**I**), with the aim of studying the interaction of PLA microfibers with the surrounding tissue. (**A**–**C**) Luxol fast blue stains (4×) of cross sections showing the distribution of PLA microfibers (marked with arrows) within the HA conduit (marked with *). The HA conduit walls present signs of incipient degradation in all experimental groups, since their porous structure presents voids and breaks in continuity (marked with circles). (**D**–**F**) Hematoxylin-eosin staining (63×) showing the distribution of the tissue and the PLA microfibers for the UMC, MMC and MMC + hSC groups, respectively. Some voids are observed between the tissue and the PLA microfibers for the UMC (**D**) and MMC (**E**) groups, whereas closer contact between the tissue and PLA microfibers is observed for the MMC + hSC group (**F**). (**G**–**I**) Luxol fast blue stains (63×) showing the interaction between PLA microfibers and myelinated nerve fibers (in dark blue), macrophages (in light blue) and collagen (in red) for the UMC, MMC and MMC + hSC groups, respectively. For the UMC (**G**) and MMC (**H**) groups, some voids are observed between myelinated nerve fibers and PLA microfibers, whereas a much closer distribution of myelinated nerve fibers and PLA microfibers without voids is appreciated for the MMC + hSC group (**I**). Scale bar = 500 µm for 4× images, 100 µm for 63× images.

**Figure 9 biomedicines-10-00963-f009:**
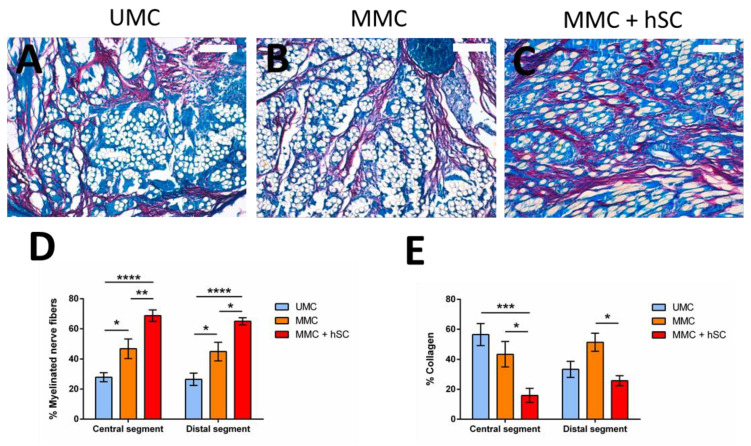
Luxol fast blue staining of tissue samples from the three experimental groups 6 months after surgery at the distal portion and quantification of the area of myelinated nerve fibers and collagen. (**A**–**C**) Luxol fast blue images of UMC, MMC and MMC + hSC, respectively, showing the collagen in red, the myelinated nerve fibers in dark blue, the macrophages/giant cells in light blue and the PLA/vessels/cavities in white. Magnification of 20×. Scale bar = 100 µm. (**D**,**E**) Quantification of the area occupied by myelinated nerve fibers and collagen, respectively, after excluding the area occupied by PLA microfibers, vessels and cavities. Statistically significant differences are indicated by *, **, *** or ****, indicating a *p*-value below 0.05, 0.01, 0.001 or 0.0001, respectively.

**Figure 10 biomedicines-10-00963-f010:**
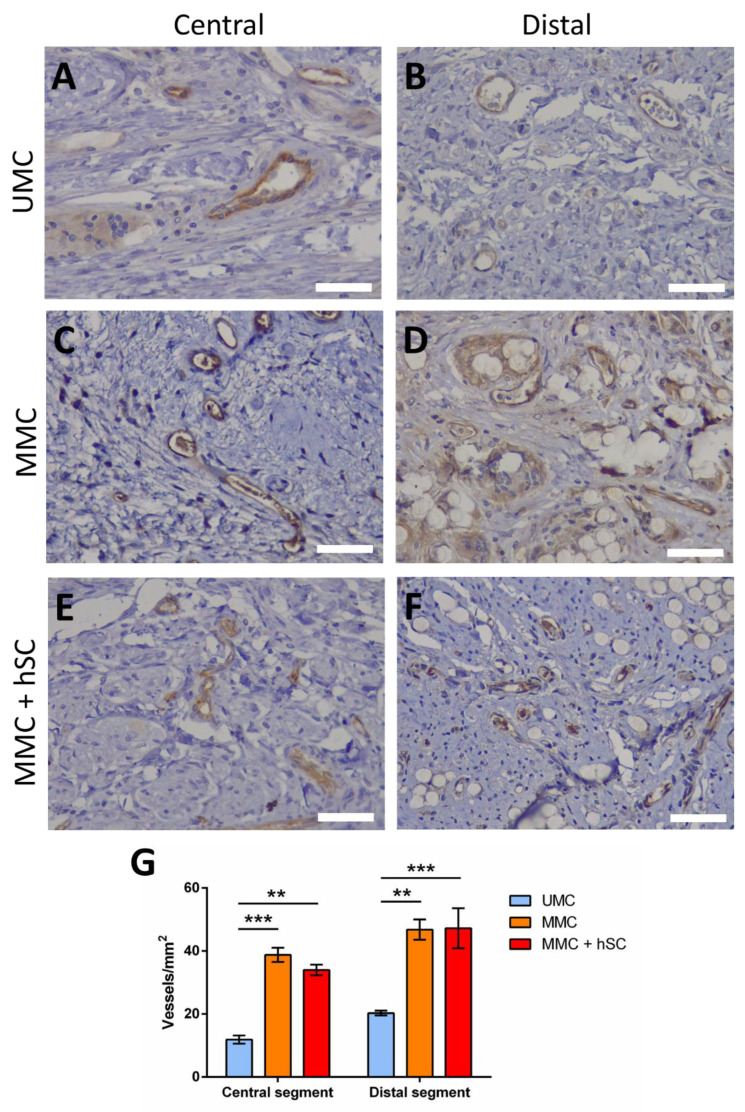
CD31 immunostaining images at central and distal portions and quantification of the number of vessels. (**A**,**B**) CD31 immunostaining of the UMC group at the central and distal portion, respectively. (**C**,**D**) CD31 immunostaining of the MMC group at the central and distal portion, respectively. (**E**,**F**) CD31 immunostaining of the MMC + hSC group at the central and distal portion, respectively. Blood vessels are marked with brown color. Scale bar = 50 µm. (**G**) Quantification of the number of vessels/mm^2^ for the three groups at the central and distal portions. Statistically significant differences are indicated by ** or ***, indicating a *p*-value below 0.01 or 0.001, respectively.

## Data Availability

All the data generated in this research are included in the manuscript.

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
