# Peer review of "Novel Tissue-Engineered Multimodular Hyaluronic Acid-Polylactic Acid Conduits for the Regeneration of Sciatic Nerve Defect"

_biomedicines, 2022, doi:10.3390/biomedicines10050963_

Round 1
Reviewer 1 Report
Dear authors,
I really appreciate your work, it was an interesting device with a lot of potential and the manuscript, except for English mistakes, is well written, nevertheless I have some concerning about it. I believe that functional tests should be performed to strengthen your results and that should be necessary add the UMC + hSC as additional experimental group.
Other considerations:
Introduction:
- Page 2: “If this gap is longer than 5 mm, the regeneration cannot be achieved naturally, and a traumatic neuroma is formed”, please remove “and a traumatic neuroma is formed”.
- Moreover, everywhere you have to change autograph with AUTOGRAFT or autologous graft. The same for allograph, that should be replaced with allograft.
- Page 3: “Furthermore, blood vessels in regenerating nerves serve as pathways over which Schwann cells (SC) can migrate to form bands of Büngner that pro-mote axonal regeneration [24].” Here bibliography should be improved.
Material and methods are well written and well detailed, with the experimental groups well described.
Results:
- Figure 3B: the multimodularity of the conduit cannot be appreciated in this picture, please replace it with a high resolute picture.
- Figure 6B and C: authors should add an arrow or an arrow head to indicate where the two moduli of the conduit are divided.
- Page 12: “The HA conduit presents a quite preserved structure, with only an incipient degradation 6 months after its implantation (Figure 7 A-C and Figure 7 G-I)”. In my opinion I did not observed an incipit of degradation looking at Figure 7, maybe authors should add quantitative data about the thickness of the conduit after 6 months of implantation or better discuss how they observed degradation since from this Figure I can’t understand it. Unfortunately, I think in general that Figure 7 didn’t give us clear information. For example, it is difficult to observe fibroblast, collagen, ecc… Other stainings (like Trichrome staining) should give you more clear information.
- Page 15: “A slight decrease was observed for the distal portion as expected since the regeneration comes from the proximal end to the distal one.” This sentence should be removed.
- Figure 9: in the Figure legend you should add information about what the staining detects with the different color.
- Page 16: “Furthermore, the MMC groups presented smaller vessels than the UMC group, indicating an increased angiogenesis.” If you state it, you should add at least one reference in which someone said that small vessels are an index of increased angiogenesis. Moreover, to complete your analysis you should add blood vessels dimension and not only the number of them.
- I really appreciate the paragraph about the study limitation, and I agree that functional recovery is really important, but for me are also necessary.
Conclusion:
“The multimodular design facilitates a more uniform distribution of cells pre-seeded in the conduit than in the case of a single, equal length module.” For which reason multimodular design should be influence cells distribution? Author should better explain this phenomenon. Please add references and better discuss this point also in the Result section 3.2.
Author Response
REVIEWER 1
Dear authors, I really appreciate your work, it was an interesting device with a lot of potential and the manuscript, except for English mistakes, is well written, nevertheless I have some concerning about it. I believe that functional tests should be performed to strengthen your results and that should be necessary add the UMC + hSC as additional experimental group.
- This is a pilot study that only seeks to have a first approximation to the performance that the multimodular approach has on the regeneration of peripheral nerves. For this reason, the study focuses on a histological analysis, rather than a functional one, with a small number of specimens (reduction principle in animal experimentation). Once the good regeneration of the MMC + hSC group has been confirmed by histological analysis, the next step will be to carry out a new trial with a larger number of animals implanted only with the MMC + hSC device compared with control group, giving a greater statistical robustness to the results and, in which, the functional analysis will be a priority. However, there is a clear relationship between the histological findings and the motor function recovery in the different stages of a sciatic nerve injury and its regeneration [1,2]. Therefore, a discussion concerning the relationship of motor function and electrophysiology alteration with the histomorphology change in peripheral nerve injury has been added to the manuscript. This extended discussion has been highlighted in yellow color.
- Unfortunately, this is a completed study that does not allow the incorporation of new experimental groups. However, we think that the chosen groups allow to observe the effect of the unimodular approach versus the multimodular approach (UMC vs MMC groups) and the effect of pre-seeded Schwann cells (MMC vs MMC + hSC groups). The inclusion of the UMC + hSC group would have given us little additional information and was not included in the study. Moreover, it was shown in vitro that the seeding and the coverage of the lumen of the HA conduit with the pre-seeded Schwann cells was better in the MMC group than in the UMC, so it was decided to pre-seed Schwann cells only in the MMC group, following the reduction principle in animal experimentation. In addition, the MMC group performed better than the UMC group in vivo and, therefore, is more convenient to introduce the Schwann cells only in the MMC group (reduction principle in animal experimentation). This aspect has been introduced in the discussion of the results in the manuscript, highlighted in yellow color.
Other considerations:
Introduction:
- Page 2: “If this gap is longer than 5 mm, the regeneration cannot be achieved naturally, and a traumatic neuroma is formed”, please remove “and a traumatic neuroma is formed”.
As suggested, the sentence “and a traumatic neuroma is formed” has been removed.
- Moreover, everywhere you have to change autograph with AUTOGRAFT or autologous graft. The same for allograph, that should be replaced with allograft.
As suggested, the word autograph has been replaced by autograft and the word allograph has been replaced by allograft. Corrections have been highlighted in yellow in the manuscript.
- Page 3: “Furthermore, blood vessels in regenerating nerves serve as pathways over which Schwann cells (SC) can migrate to form bands of Büngner that pro-mote axonal regeneration [24].” Here bibliography should be improved.
As suggested, the bibliography that supports this sentence has been improved. Corrections have been highlighted in yellow in the manuscript.
Material and methods are well written and well detailed, with the experimental groups well described.
Results:
- Figure 3B: the multimodularity of the conduit cannot be appreciated in this picture, please replace it with a high resolute picture.
The gap between modules has a very small length, around 100 µm. For this reason, it is difficult to distinguish the multimodularity of the conduit with a macroscopic image such as Figure 3B. The aim of Figure 3B is to give a macroscopic overview of the whole device, with all its components arranged in the final disposition. However, to better visualize the multimodularity, an insert with an amplified image of the gap between HA modules has been introduced in Figure 3B to better visualize its dimension. Modifications to the figure legend are highlighted in yellow color in the manuscript.
- Figure 6B and C: authors should add an arrow or an arrow head to indicate where the two moduli of the conduit are divided.
As suggested, an arrow has been added to indicate the separation between the two modules. Corrections in the figure legend have been highlighted in yellow in the manuscript.
- Page 12: “The HA conduit presents a quite preserved structure, with only an incipient degradation 6 months after its implantation (Figure 7 A-C and Figure 7 G-I)”. In my opinion, I did not observe an incipit of degradation looking at Figure 7, maybe the authors should add quantitative data about the thickness of the conduit after 6 months of implantation or better discuss how they observed degradation since from this Figure I can’t understand it. Unfortunately, I think in general that Figure 7 didn’t give us clear information. For example, it is difficult to observe fibroblast, collagen, ecc… Other stainings (like Trichrome staining) should give you more clear information.
The incipient degradation of the HA conduit is based on the detailed visualization of histological images, where the HA porous structure presents voids and breaks in continuity for all the experimental groups. For a better visualization of these voids, they have been marked with circles in Figure 7 and Figure 8. The discussion about how the degradation of the HA conduit is observed has been extended in the manuscript (highlighted in yellow color).
- Page 15: “A slight decrease was observed for the distal portion as expected since the regeneration comes from the proximal end to the distal one.” This sentence should be removed.
As suggested, this sentence has been removed.
- Figure 9: in the Figure legend you should add information about what the staining detects with the different color.
As suggested, the legend of Figure 9 has been extended, adding more information about what the staining detects with the different colors (highlighted in yellow in the manuscript).
- Page 16: “Furthermore, the MMC groups presented smaller vessels than the UMC group, indicating an increased angiogenesis.” If you state it, you should add at least one reference in which someone said that small vessels are an index of increased angiogenesis. Moreover, to complete your analysis you should add blood vessels dimension and not only the number of them.
This sentence has been replaced by the following one: “Therefore, the MMC groups presented more microvessels which correlate with an increased expression of angiogenic markers as reported previously [3]”. This correction is highlighted in yellow color in the manuscript.
- I really appreciate the paragraph about the study limitation, and I agree that functional recovery is really important, but for me are also necessary.
As stated in the paragraph about the study limitations, functional recovery was not assessed by electrodiagnostic studies. As previously discussed, it is a pilot study with a small number of animals and the regeneration was studied only by histomorphology assessment. However, there is a clear relationship between the histological findings and the motor function recovery in the regeneration of the sciatic nerve [1,2], and a discussion concerning the relationship of motor function and electrophysiology alteration with the histomorphology change in peripheral nerve injury and regeneration has been added to the discussion of the results. This extended discussion has been highlighted in yellow in the manuscript.
Conclusion:
“The multimodular design facilitates a more uniform distribution of cells pre-seeded in the conduit than in the case of a single, equal length module.” For which reason multimodular design should be influence cells distribution? Author should better explain this phenomenon. Please add references and better discuss this point also in the Result section 3.2.
The multimodular design entails a more uniform distribution of the pre-seeded Schwann cells because it facilitates the seeding process: the cells are seeded at both ends of each module, so a shorter module reduces the distance between the seeding zones and, thus, better coverage of the lumen of the conduit is achieved. In addition, the use of shorter modules allows better exchange of nutrients than the use of a longer module, whose center has fewer nutrients available. This can also affect the cell distribution in the central part of the lumen of the conduit. This explanation has been included in the discussion of the results of section 3.2, adding some references, as suggested.
REFERENCES
- Yamasaki, T.; Fujiwara, H.; Oda, R.; Mikami, Y.; Ikeda, T.; Nagae, M.; Shirai, T.; Morisaki, S.; Ikoma, K.; Masugi-Tokita, M.; et al. In vivo evaluation of rabbit sciatic nerve regeneration with diffusion tensor imaging (DTI): Correlations with histology and behavior. Magn. Reson. Imaging 2015, 33, 95–101, doi:10.1016/j.mri.2014.09.005.
- Bendszus, M.; Wessig, C.; Solymosi, L.; Reiners, K.; Koltzenburg, M. MRI of peripheral nerve degeneration and regeneration: Correlation with electrophysiology and histology. Exp. Neurol. 2004, 188, 171–177, doi:10.1016/j.expneurol.2004.03.025.
- Ruiz-Saurí, A.; Valencia-Villa, G.; Romanenko, A.; Pérez, J.; García, R.; García, H.; Benavent, J.; Sancho-Tello, M.; Carda, C.; Llombart-Bosch, A. Influence of Exposure to Chronic Persistent Low-Dose Ionizing Radiation on the Tumor Biology of Clear-Cell Renal-Cell Carcinoma. An Immunohistochemical and Morphometric Study of Angiogenesis and Vascular Related Factors. Pathol. Oncol. Res. 2016, 22, 807–815, doi:10.1007/s12253-016-0072-7.

Reviewer 2 Report
This is a very interesting study concerning a tissue-engineered multimodular nerve guidance conduit for the treatment of large nerve damages by a polylactic acid (PLA) microfibrillar structure inserted inside to several co-linear hyaluronic acid (HA) conduits. The result showed that the multimodular approach contributed to a better vascularization through the micrometrical gaps between HA conduits, while the pre-seeded Schwann cells increased axonal growth.
This is a well designed study. However, the introduction is too long and wordy. The authors should it shorter and brief.
This study was lack of motor function and electrophysiology assessment, which ere the essential parameters to measure the functional recovery, not merely based on the histomorphology assessment in this study. The author at least should make a brief discussion concerning the relationship of motor function and electrophysiology alteration related to histomorphology change in peripheral nerve injury.
Author Response
REVIEWER 2
This is a very interesting study concerning a tissue-engineered multimodular nerve guidance conduit for the treatment of large nerve damages by a polylactic acid (PLA) microfibrillar structure inserted inside to several co-linear hyaluronic acid (HA) conduits. The result showed that the multimodular approach contributed to a better vascularization through the micrometrical gaps between HA conduits, while the pre-seeded Schwann cells increased axonal growth.
This is a well-designed study. However, the introduction is too long and wordy. The authors should it shorter and brief.
As suggested, the length of the introduction has been reduced. It has been rewritten and its total length has been reduced in 33 lines.
This study was lack of motor function and electrophysiology assessment, which ere the essential parameters to measure the functional recovery, not merely based on the histomorphology assessment in this study. The author at least should make a brief discussion concerning the relationship of motor function and electrophysiology alteration related to histomorphology change in peripheral nerve injury.
As suggested, a discussion concerning the relationship of motor function and electrophysiology alteration with the histomorphology change in peripheral nerve injury and its regeneration has been added to the discussion of the results. This extended discussion has been highlighted in yellow in the manuscript.

Round 2
Reviewer 1 Report
The manuscript quality has been improved and authors reply to almost all my requests.